# Evaluation of postural therapy using lateral position according to fetal back orientation on breech presentation and breech recurrence (BRLT study): An open-label randomized controlled trial

**Hiroki Shinmura[1]\*, Youhei Tsunoda[2], Takashi Matsushima[1], Ryuhei Kurashina[1], Asako Watanabe[1], Eika Harigane[1], Nozomi Ouchi[1], Shunji Suzuki[2]**

**1** Department of Obstetrics and Gynecology, Nippon Medical School Musashikosugi Hospital, Kawasaki, Kanagawa, Japan, **2** Department of Obstetrics and Gynecology, Nippon Medical School Hospital, Bunkyo, Tokyo, Japan

\* h-shimmura@nms.ac.jp

## Abstract

### Background

In Japan, the lateral position method is known as a postural therapy for breech presentation wherein the mother lies down in lateral position according to the orientation of the fetal back. Few studies have formally tested lateral position management for breech presentation, and no method exists to prevent breech recurrence after cephalic version. We hypothesized that postural management comprising a combination of opposite-side lateral position for breech presentation and same-side lateral position after cephalic version demonstrates a clinically relevant effect size on breech presentation.

### Methods and findings

We conducted a stratified, open-label randomized controlled trial at an academic hospital in Kawasaki, Japan. A total of 200 women diagnosed with breech presentation between 28 + 0 and 30 + 0 gestational weeks were randomized to postural management (*n* = 100) or control (no intervention, *n* = 100) group. The intervention was instruction every 2 weeks on lying in the lateral position on the opposite-side of fetal back for breech presentation and on the same-side of fetal back for head-first presentation. The primary outcome was the rate of fetuses in breech presentation at 37 weeks of gestation, and the secondary outcomes were cesarean delivery, cesarean delivery for breech presentation, head presentation 2, 4, and 6 weeks later, breech presentation recurrence, and adverse events. Breech presentation rate at 37 gestational weeks was 11% in the intervention group, using the combination of the opposite-side and same-side lateral positions, compared with 19% in the control group. However, we found no statistical significance in the intention-to-treat analysis (11% [11/100] versus 19% [19/100]; relative risk, 0.58 [95% CI, 0.29 to 1.15]; *p*

**Data availability statement:** All data supporting the findings of this study are stored in the manuscript and the Supporting information files.

**Funding:** The author(s) received no specific funding for this work.

**Competing interests:** The authors declare no conflicts of interest.

= 0.11). In the control group, 23 participants (23%) unknowingly took the same posture as the intervention group, and the prespecified per-protocol analysis excluding crossover found the same direction of effect but with statistical significance. In the intention-to-treat analysis, the intervention group had a higher cephalic version rate 2 weeks after the instruction (69% [69/100] versus 54% [54/100]; relative risk, 0.67 [95% CI, 0.47 to 0.96]; $p$ = 0.029), and lower breech presentation recurrence rates (2% [2/91] versus 10% [9/88]; relative risk, 0.22 [95% CI, 0.048 to 0.97]; $p$ = 0.031) than the control group. Regarding adverse events in the intervention group, three participants experienced discomfort and one participant complained of pain in the lateral abdomen; these symptoms resolved spontaneously.

## Conclusions

For breech presentation at the beginning of the third trimester, providing postural therapy instruction on opposite-side lateral positioning and same-side lateral positioning was associated with 8% reduction of breech fetuses at 37 gestational weeks compared with the control group, but this primary endpoint did not reach statistical significance. Regarding the secondary endpoints, the intervention group showed a significantly higher rate of cephalic version after 2 weeks and lower rate of breech recurrence. The direction of the effect of postural therapy based on fetal back position on breech presentation was promising, and further research to validate this approach, with consideration for unplanned participant crossover, may be warranted.

## Trial registration

UMIN Clinical Trials Registry (UMIN000043613, https://center6.umin.ac.jp/cgi-open-bin/ctr_e/ctr_view.cgi?recptno=R000049800).

---

Author summary
### Why was this study done?

- In Japan, the lateral position method has been practiced as a postural therapy for breech presentation; however, no randomized controlled trials have been conducted to confirm this effect.

- Theoretically, the same side lateral position method, a modification of the opposite side lateral position method previously known in Japan, can prevent breech presentation recurrence after cephalic version; however, to our knowledge, this has never been demonstrated in a study.

### What did the researchers do and find?

- We conducted a stratified, open-label randomized controlled trial to examine the effect of lateral position management for breech presentation at an academic hospital in Kawasaki, Japan.

- A total of 200 pregnant women whose fetuses were in breech presentation at 28–30 gestational weeks were randomly assigned to the lateral position management (intervention)

group (100 participants) or the control group with no intervention (100 participants), and the intervention group was then instructed to lie on their side according to fetal back, as shown in a separate figure (Fig 1) every 2 weeks.

- At 37 weeks of pregnancy, breech babies in the intervention group were about half of those in the control group, but this result was not statistically significant.

## What do these findings mean?

- This study demonstrated the potential of a new concept in breech presentation management, just by lying down in lateral position based on the fetal back.

- Although further study is needed, the opposite side lateral position may potentially reduce breech babies and the same side lateral position may potentially prevent breech presentation recurrence.

## Introduction

Breech presentation occurs in 3%–4% of all term pregnancies, and vaginal delivery in breech presentation is associated with perinatal morbidity and mortality compared with planned cesarean section [1]. As a large randomized controlled trial published by Hannah and colleagues (2000), cases of cesarean section for breech presentation have increased worldwide [1–5]. Conversely, a cohort study showed that under the guidance of experienced physicians, vaginal delivery in breech presentation is as safe as cesarean sections, and there is a movement to rethink vaginal delivery in breech presentation [6]. In either case, cephalic version is important because it may avoid the risks of cesarean section and vaginal delivery for breech presentation. External cephalic version is an established treatment method for breech presentation, which is performed after 36 gestational weeks, but it poses problems such as pain and unsuccessful cases [7–10]. Conversely, treatment for breech presentation before 36 weeks has not yet been established. Postural therapies, such as knee–chest position and deep

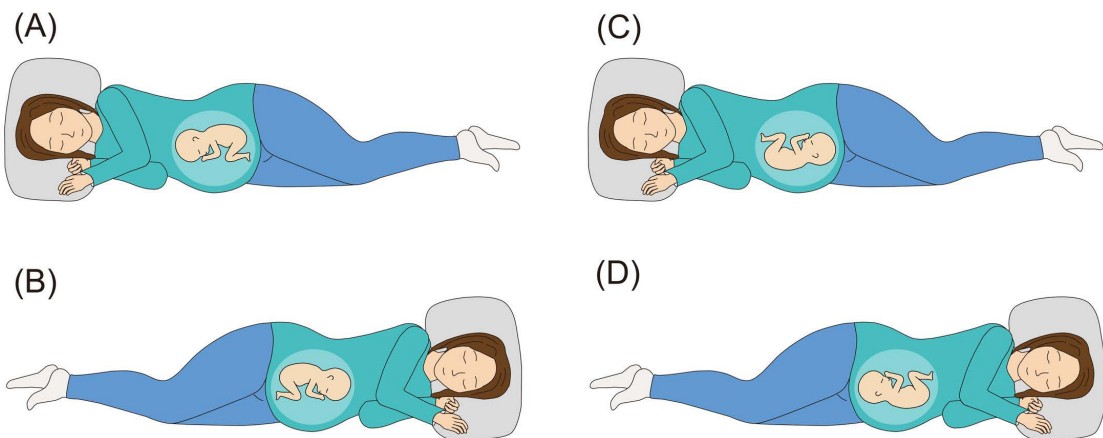

**Fig 1. Opposite side lateral position for breech presentation:** (A) the right lateral recumbent position is taken if the fetal back is on the mother's left side. (B) The left lateral recumbent position is taken if the fetal back is on the mother's right side. **Same side lateral position after cephalic version:** (C) the right lateral recumbent position is taken if the fetal back is on the mother's right side. (D) The left lateral recumbent position is taken if the fetal back is on the mother's left side.

Trendelenburg position, or acupuncture and moxibustion are known treatments that can be performed before 36 weeks; however, evidence to recommend these treatment remains insufficient [11–19].

Wigand (1812) described the lateral position as effective when correcting oblique position to cephalic position [20]. The lateral position on the opposite side of fetal back for breech presentation (Fig 1, opposite side lateral position, A and B), which is a modification of Wigand's method, been performed along with the knee–chest position in Japan since 1943, when this report was published by Taoka (1943) [21,22]. Our previous cohort study demonstrated its possible usefulness; however, the small effect size of the single lateral position instruction remained an issue in the study [23]. Considering the absence of complications observed in the study and a meta-analysis showed that sleeping in lateral position is safe, long-term lateral position performing to compensate for its small effect size was considered acceptable [24]. Currently, no method exists to prevent breech presentation recurrence after cephalic version. Its effectiveness had never been verified, but the lateral position on the same side of fetal back (Fig 1, same side lateral position, C and D) was theoretically a safe method for preventing breech presentation recurrence after cephalic version [24].

Hence, we hypothesized that a combination of multiple sessions of the opposite side lateral position instruction every 2 weeks and the same side lateral position method after cephalic version would demonstrate a clinically relevant effect size on breech presentation. In this study, we conducted the largest randomized controlled trial of postural therapy for breech presentation to determine whether serial instructions of the opposite side lateral or same side lateral positions can reduce term breech presentation cases.

## Methods

### Study design and participants

We performed a stratified, open-label randomized controlled trial with two parallel groups allocated in a 1:1 ratio to evaluate lateral position management for breech presentation, in comparison with the usual prenatal management care at an academic hospital in Kawasaki, Japan.

The study protocol was designed according to the Standard Protocol Items: Recommendations for Interventional Trials (SPIRIT) 2013, with approval of the Nippon Medical School Musashikosugi Hospital Ethics Review Board on March 9, 2021 (approval number: 599-2-69) [25]. This trial was registered in the UMIN Clinical Trials Registry (UMIN000043613) on March 15, 2021, and participant recruitment began on April 1, 2021. The first participant was enrolled in the trial on April 2, 2021 and the last participant on November 18, 2023. In reporting findings, this study conformed to the CONSORT (Consolidated Standards of Reporting Trials) 2010 statement [26]. The study protocol was amended on July 7, 2022 (within the study period), based on the external reviewer's opinion. The protocol has been published in a peer-reviewed open-access journal [27]. Notification of the study was made through the hospital website, with no advertising used. Pregnant women who presented for a prenatal checkup at 28–30 gestational weeks, where routine ultrasonographic screening was performed, and exhibited a breech presentation were evaluated by their attending obstetricians for eligibility for the trial. We set 28–30 gestational weeks as the start of this study to begin the intervention by 32 gestational weeks, when amniotic fluid volume is generally at its maximum and the fetus may rapidly become less rotatable after that point [28].

Inclusion criteria were as follows: gestational age between 28 + 0 and 30 + 0 gestational weeks, pregnant women undergoing prenatal checkups at our hospital, and breech presentation diagnosed by ultrasonography. Exclusion criteria were as follows: less than 20 years

old (considering the social reason that the consent of parent or guardian was also required in Japan), scheduled cesarean delivery (included placenta previa, cesarean delivery history, and myomectomy history), scheduled delivery at another hospital, multiple pregnancy, transverse position, any treatment already in place for breech presentation or complications with difficulty in postural management (included cardiovascular disease).

The participants' eligibility was evaluated on the same day as the diagnosis of breech presentation, and eligible women were recruited to the study by their obstetricians on the same day. We used preprepared documents to prevent variation in recruiters explaining the trial. All participants provided written informed consent before randomization.

## Randomization and masking

The allocation tables were computer-generated by a researcher independent of those who conducted the eligibility assessment and recruitment. Random block sizes of 4, 6, and 8 were used. Considering that primiparous women were less likely to rotate their babies than multiparous women, stratified randomization was conducted by parity (primiparous or multiparous), and two allocation lists were used (one for primiparous women and one for multiparous women) [29,30]. A consecutively numbered allocation list generated by the computer was sealed by security void stickers. If the sticker was removed, it would leave the word "Void" on the list. Moreover, sealing stickers were put on the back of the list to prevent the allocation from being seen through. Then a nurse at the outpatient section of obstetrics performed the randomization by removing the stickers from the list, who was independent of the investigators responsible for enrollment. Group assignment was immediately communicated to the researchers and participants, with no masking. However, the group allocation was masked to the statistical analyst who conducted the analysis.

## Procedures

Opposite side lateral position in this study was defined as follows: if the fetal back was on the mother's left side, the mother took the lateral recumbent position with her right side down; if the fetal back was on the mother's right side, the mother took the lateral recumbent position with her left side down (Fig 1, opposite side lateral position, A and B). Fetal back position was verified by ultrasonography along with estimated fetal weight and amniotic fluid volume at checkup every 2 weeks, and which side the mother should lie on was also updated every 2 weeks. Women assigned to the intervention group were instructed to take the lateral position three times a day for 15 min each time for 2 weeks. They were allowed to remain in the lateral position for more than 15 min as long as no physical problems occurred, and each 15 min was counted as one session. For example, if they could maintain the lateral position for more than 45 min on a given day, they were considered to have performed at least three sessions, and that day was regarded as a successful day. Obstetricians who guided the women in the intervention group were pretrained before trial so that they could correctly instruct the women on which side they should sleep on. Opposite side lateral position instruction was given at checkup every 2 weeks until the pelvic position was changed to the head-first position. After the cephalic version, we instructed the intervention group to perform the same side lateral position.

Herein, same side lateral position in this study was defined as follows: if the fetal back was on the mother's right side, the mother took the lateral recumbent position with her right side down; if the fetal back was on the mother's left side, the mother took the lateral recumbent position with her left side down (Fig 1, same side lateral position, C and D). The fetal presentation was confirmed at bi-weekly checkups, and the participant continued to be instructed on

the opposite side lateral position if in breech presentation, and the same side lateral position if in head-first presentation until delivery. If the fetus was in the transverse position during the trial, the intervention group was instructed as follows: if the fetal head was on the mother's left side, the mother took the lateral recumbent position with her right side down; if the fetal head was on the mother's right side, the mother took the lateral recumbent position with her left side down. We did not recommend the knee–chest position and other treatments for breech presentation. Furthermore, participants were instructed to discontinue the lateral position if they felt uncomfortable during the designated posture and to report the incident. We asked them to fill out a daily position record to confirm that they performed the proper posture. Perinatal care other than the intervention was performed according to the 2020 guidelines for obstetrical practice in Japan.

Likewise, the control group received perinatal care based on the 2020 guidelines for obstetrical practice in Japan. Any other treatments for breech presentation before 36 gestational weeks were not recommended. To reassure women in the control group and prevent them from deviating from the protocol, the researchers explained that postural, acupuncture and moxibustion therapies were not recommended because of lack of evidence, and literature suggested that even if the breech presentation was present at the time of allocation, approximately 80% will return to the head-first presentation by 37 gestational weeks. The control group was not prohibited from taking the lateral position, although the proper side was not explained deliberately to avoid mimicking the theoretically correct posture of the intervention group. The control group was also asked to complete a daily position record form until delivery to confirm the type of posture it took.

If the breech presentation persisted at term, all participants were offered the choice of a scheduled C-section, vaginal delivery or external cephalic version. Cesarean section was scheduled at 38 gestational weeks. We informed all participants in advance that they could withdraw from the trial at any time and for any reason. To avoid possible deviations from the study, written instructions were provided in advance for several anticipated questions and inquiries during the study.

## Outcomes

The primary endpoint was the percentage of breech presentation cases at 37 gestational weeks. The secondary outcomes were cesarean delivery, cesarean delivery for breech presentation as the primary reason, head presentation 2, 4, and 6 weeks later, breech presentation recurrence after cephalic version, and adverse events. The presentation at each time point was confirmed by ultrasonography each time by the investigators in charge of the antenatal checkup that day. The primary and all secondary endpoints, excluding adverse events, were evaluated by comparing the percentage of fetuses with breech presentation between the intervention and control groups. Only breech presentation recurrence was compared between groups converted to cephalic presentation at least once. To evaluate perinatal prognosis in relation to adverse events, we examined participants' gestational age at delivery, delivery mode, birth weight, Apgar scores at 1 and 5 min, umbilical artery pH at birth, bleeding amount during delivery, and perinatal complications were also collected. To prevent missing data, we regularly conducted study progress-sharing sessions wherein researchers shared information about items that could easily be missing.

We changed the primary outcome and sample size during the study period because when the protocol paper was peer-reviewed, a reviewer strongly recommended that the cephalic version rate after 2 weeks (the original primary endpoint) be clinically unimportant and that the clinically important breech presentation at term (the secondary endpoint initially) be the primary endpoint. We discussed and carefully considered this recommendation. A power

analysis of the case with the breech presentation at term as the primary endpoint showed that the sample size increase was within the feasible range. Thus, we decided to change the primary endpoint and the sample size, with the approval from the ethics review board. We also changed the details in the trial registry.

## Sample size

The sample size was calculated on the basis of our prior cohort study on the lateral position for breech presentation and other previous studies. In our prior cohort study, the cephalic conversion rate at term in the intervention group was 94% (16/17) [23]. Hence, we applied the same rate in the intervention group of the current study. In the control group, the previous rate was 77% (30/39). A study dealing with breech presentation at 28–30 weeks, which is the same gestational age as our study, showed a cephalic conversion rate of 75% (non-cephalic position of 21.4% at 28–30 weeks spontaneously decreased to 5.3% after 37 weeks) [31]. In other studies, the cephalic conversion rate was 84% for breech presentation at less than 28 weeks, but in our study, it was estimated to be lower than 84% because this study was conducted after 28 weeks [32,33]. Therefore, we selected 80% as the cephalic conversion rate for the control group in the current study. A power analysis was performed so that the 14% absolute risk difference (94% versus 80%) could be detected with 80% power at an $\alpha$-value of 0.05 (two-sided) in the $\chi^2$ test, with 182 as the required sample size. Effect sizes were calculated using G*Power (version 3.1, Faul, Erdfelder, Lang, and Buchner, Düsseldorf, Germany). We set 100 participants per arm, totaling 200 women, considering the missing value of approximately 10%.

## Statistical analysis

We assessed the outcomes through intention-to-treat and per-protocol analyses according to a predesigned protocol and analysis plan. The primary and secondary outcomes were two-category data, and the intervention and control groups were compared using the $\chi^2$ test (Pearson's test or Fisher's test). Unadjusted relative risks were calculated as the main result with 95% confidence intervals. We also calculated the absolute risk reductions. Furthermore, we employed Student $t$ test for normally distributed continuous variables, and the Mann–Whitney $U$-test for nonnormally distributed ones. Subgroup analyses were separately performed for primiparous participants and multiparous participants. The intention-to-treat analysis included protocol deviations and compared all participants' results between the two groups. Participants were considered to have successfully performed the lateral position management if they had performed it on the correct side for at least 45 min in at least half of the 2-week period, and whether the patient was able to perform the lateral postural therapy as per protocol was recorded in the categorical variable.

Considering the nature of the lateral position, we could not prohibit the control group from executing lateral position; therefore, many participants in the control group would probabilistically be in the same posture as the intervention group. Already in the design phase of this study, the first randomized controlled trial to examine the lateral position management which has never been tested with a high level of evidence, we considered that the presence of a potential lateral position group in the control group, as described above, would be a barrier to answering the pure question of whether lying on one's side is really enough to achieve cephalic version. Therefore, a per-protocol analysis that excluded the potential intervention groups within control group was necessary to verify the true efficacy of the lateral position, as distinct from the clinical effectiveness of the intention-to-treat analysis. The per-protocol analysis compared the intervention group, which excluded participants with a success rate of

less than 50%, with a control group, which excluded participants with a rate of correct lateral positioning greater than 50% (participants who had performed almost the same therapy as the intervention group). A risk of attrition bias is possible; however, the analysis was conducted strictly according to the protocol as previously decided.

For missing values, we used mean imputation for continuous variables and pairwise deletion for categorical variables and reported the results when used. A *P*-value <0.05 was considered statistically significant, and two-sided tests were conducted for all analyses. All statistical data were analyzed using IBM SPSS Statistics (version 29.0, IBM Corp.).

### Patient and public involvement

Patients and the public were not involved in the design, implementation, or dissemination of this study.

## Results

Between April 2, 2021 and November 18, 2023, 421 pregnant women who presented for a 28–30 weeks obstetric checkup and had a breech presentation on ultrasonography were evaluated for eligibility in person. Of those, 165 women were ineligible, 256 women met the eligibility criteria (Fig 2). Because of refusal to participate, no approach for recruitment or other reasons, 55 women were not included. Of the 201 participants enrolled in this study, one participant has had a history of cesarean section and was excluded before allocation. Ultimately, 200 participants were randomly assigned, 100 to the intervention group and 100 to the control group. Six of the total participants delivered at other hospitals, but information was provided by the other hospitals and the participants whenever possible. Follow-up was conducted until February 6, 2024, when the outcomes of all 200 patients were confirmed. In the intervention group, two were instructed in the incorrect side and eight obtained a success rate of less than 50% in performing the lateral position. In the controls, one received the same instruction as the intervention group just before the cephalic version, three confessed to performing the same lateral positioning technique as the intervention group, and 23 had a success rate of greater than 50% in lateral positioning in the same side as the intervention group by the time of the cephalic version. In addition, one of the controls have had a previous cesarean section after allocation, and a scheduled cesarean section was performed at 38 weeks. All 200 participants were included in the intention-to-treat analysis. The per-protocol analysis included 90 participants in the intervention group and 73 in the control group.

Table 1 shows the characteristics of the intervention group, control group and participants excluded from the per-protocol analysis for attrition bias evaluation. The median number of gestational weeks at randomization was 28 weeks for all groups. Incomplete breech was rare in any groups at baseline. Approximately 40% of all participants had undergone fertility treatment to conceive. None of the characteristics showed noticeable differences between the intervention and control groups. Participants excluded from the per-protocol analysis tended to have a higher number of primiparous women (73% versus 62%) and fewer multiparous women (27% versus 38%) than controls, but no significant differences were noted in any of the characteristics. None of the groups showed abnormalities in amniotic fluid volume.

### Intention-to-treat analysis

As the primary outcome in the intention-to-treat analysis, breech presentation occurred at 37 weeks in 11 participants (11% [11/100]) from the intervention group and 19 (19% [19/100]) from the control group; thus, breech presentation occurred in almost a half of those in the intervention group. However, the difference was not statistically significant (absolute risk

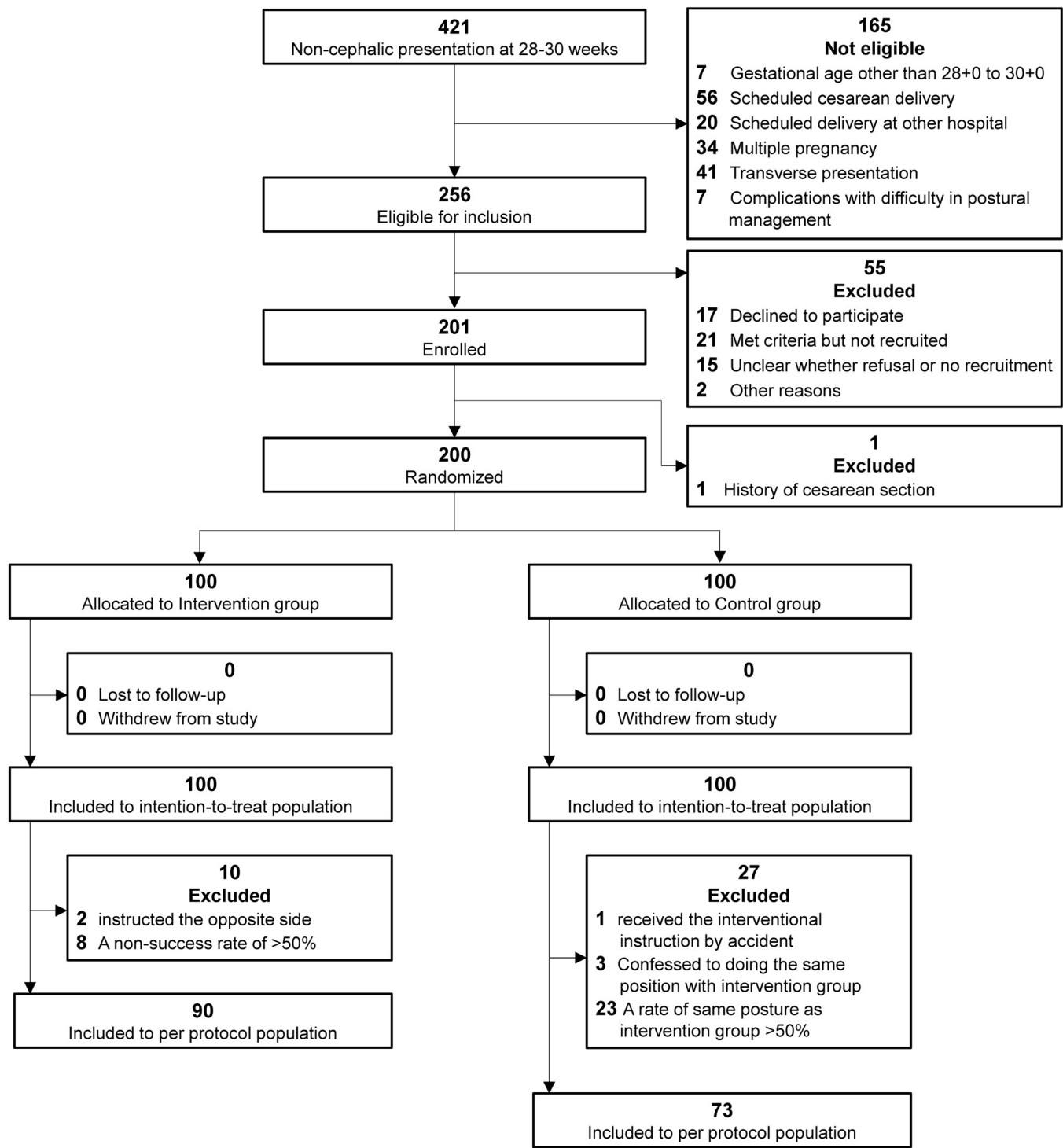

**Fig 2. Flowchart of participant screening, enrollment, randomization, and inclusion in the analysis.**

difference, −8.0% [95% CI, −17.8% to 1.84%]; relative risk, 0.58 [95% CI, 0.29 to 1.15]; $p$ = 0.11; Table 2). For four participants in the intervention group and six in the control group who delivered prematurely, fetal presentation at delivery was considered as the primary

**Table 1. Baseline characteristics of participants.**

| Characteristics | Lateral position group ($n = 100$) | Control group ($n = 100$) | Excluded from Per-protocol analysis ($n = 37$) |
|---|---|---|---|
| Gestational age at randomization(weeks+days)* | 28 + 6 (28 + 3–29 + 3) | 28 + 6 (28 + 2–29 + 2) | 28 + 5 (28 + 2–29 + 3) |
| Maternal age (years)* | 34 (31–37) | 35 (31–39) | 33 (30–36) |
| Amniotic pocket (cm)* | 3.5 (3.4–4.2) | 3.5 (3.3–4.2) | 3.7 (3.4–4.0) |
| Pre-pregnancy body mass index* | 21.6 (20.3–23.6) | 21.8 (19.9–24.2) | 22.1 (19.7–26.0) |
| Height less than 150 cm | 2 (2) | 3 (3) | 1 (3) |
| Active smoking | 1 (1) | 0 (0) | 1 (3) |
| Fertility treatment | 36 (36) | 41 (41) | 15 (41) |
| Type of breech presentation: | | | |
| Frank | 50 (51) | 46 (46) | 17 (46) |
| Complete | 48 (48) | 51 (52) | 20 (54) |
| Incomplete | 1 (1) | 2 (2) | 0 (0) |
| Position of the fetal back in the mother's body at randomization: | | | |
| Left | 48 (48) | 59 (59) | 19 (51) |
| Right | 52 (52) | 41 (41) | 18 (49) |
| Parity: | | | |
| Nulliparous | 64 (64) | 62 (62) | 27 (73) |
| Multiparous | 36 (36) | 38 (38) | 10 (27) |
| Ethnicity: | | | |
| Asian | 100 (100) | 100 (100) | 37 (100) |
| Other | 0 (0) | 0 (0) | 0 (0) |
| Location of placenta: | | | |
| Anterior | 46 (46) | 46 (46) | 22 (59) |
| Posterior | 43 (43) | 45 (45) | 11 (30) |
| Lateral | 2 (2) | 1 (1) | 0 (0) |
| Fundal | 9 (9) | 7 (7) | 4 (11) |
| Low-lying | 0 (0) | 1 (1) | 0 (0) |
| Pre-existing conditions: | | | |
| Myomas | 11 (11) | 8 (8) | 4 (11) |
| Uterine malformation | 1 (1) | 1 (1) | 0 (0) |
| Threatened premature labor | 1 (1) | 1 (1) | 0 (0) |
| Extremes of amniotic fluid volume | 0 (0) | 0 (0) | 0 (0) |
| Intrauterine growth restriction | 1 (1) | 1 (1) | 0 (0) |
| Large for gestational age | 1 (1) | 1 (1) | 0 (0) |
| Fetal anomaly | 1 (1) | 1 (1) | 0 (0) |
| Ovarian swelling | 4 (4) | 2 (2) | 1 (3) |
| History of abdominal surgery | 3 (3) | 4 (4) | 3 (8) |
| Hypertensive disorder | 3 (3) | 5 (5) | 1 (3) |
| Glucose intolerance | 6 (6) | 7 (7) | 4 (11) |
| Asthma | 5 (5) | 3 (3) | 3 (8) |
| Previous infection with COVID-19 | 3 (3) | 3 (3) | 2 (5) |
| Psychiatric disorder | 3 (3) | 2 (2) | 1 (3) |

Missing data: Lateral position group; type of breech presentation ($n = 1$), amniotic pocket ($n = 9$). Control group; height ($n = 1$), type of breech presentation ($n = 1$), amniotic pocket ($n = 8$).

Values are numbers (percentages) unless stated otherwise.

There were no significant differences between the groups in any of the characteristics listed here.

*Values are median (interquartile range).

outcome. Two weeks after the start of the positioning instruction, breech presentation was converted into cephalic in 69% (69/100) of the participants in the intervention group; this percentage was significantly more than that in the control group (54% [54/100]) (absolute risk difference, −15.0% [95% CI, −28.3% to −1.67%]); relative risk, 0.67 [95% confidence interval 0.47 to 0.99]; $p = 0.029$). After 4 or 6 weeks of instruction, the cephalic version rate did not significantly differ between such groups. After cephalic version, the intervention significantly prevented breech presentation recurrence more than the control (2% [2/91] versus Ten% [9/88]; absolute risk difference, −8.0% [95% CI, −15.0% to −1.02%]; relative risk, 0.22 [95% CI, 0.048 to 0.97]; $p = 0.031$). Participants who had non-cephalic presentation at 37 weeks were offered scheduled cesarean section, external cephalic version, or vaginal delivery, and all chose to undergo a scheduled cesarean section. No women wanted to undergo external cephalic version or vaginal delivery, and all who were non-cephalic at 37 weeks underwent a cesarean section at 38 weeks. Moreover, the overall cesarean section rate was significantly lower in the intervention group than in the control group (23% [23/100] versus 36% [36/100]; absolute risk difference, −13.0% (95% CI, −25.5% to −0.49%); relative risk, 0.64 (95% CI, 0.41 to 0.99); $p = 0.044$), although this result included one patient with a previous cesarean section.

## Per-protocol analysis

Using a predetermined method, the per-protocol analysis revealed that the intervention group had a statistically significant reduction in pelvic position at 37 weeks (9% [8/90] versus 21% [15/73]); absolute risk difference, −11.7% [95% CI, −22.6% to −0.68%]; relative risk, 0.43 [95% CI, 0.19 to 0.96]; $p = 0.020$; Table 3). Significant differences were also observed in the rate of cephalic presentation at 2 weeks later, total cesarean section rate, cesarean section rate with breech presentation as the indication, and prevention of breech presentation recurrence, but no significant differences were observed in the rate of cephalic version at 4 and 6 weeks later.

**Table 2. Primary and secondary outcomes in the intention-to-treat analysis.**

| Outcomes | No (%) of participants | | Relative risk (95% CI) | P-value |
|---|---|---|---|---|
| | **Lateral position group** | **Control group** | | |
| | ($n = 100$) | ($n = 100$) | | |
| **Primary outcome** | | | | |
| Breech presentation at 37 weeks | 11 (11) | 19 (19) | 0.58 (0.29 to 1.15) | 0.11 |
| **Secondary outcomes** | | | | |
| Cephalic version 2 weeks after the start of the intervention | 69 (69) | 54 (54) | 0.67 (0.47–0.96) | 0.029 |
| Cephalic version 4 weeks after the start of the intervention | 79 (79) | 72 (72) | 0.75 (0.46–1.23) | 0.25 |
| Cephalic version 6 weeks after the start of the intervention | 81 (81) | 77 (77) | 0.83 (0.48–1.42) | 0.49 |
| Recurrence of breech presentation after cephalic version | 2/91 (2) | 9/88 (10) | 0.22 (0.048–0.97) | 0.031* |
| Overall cesarean delivery | 23 (23) | 36 (36) | 0.64 (0.41–0.99) | 0.044 |
| Cesarean delivery due to breech presentation | 9 (9) | 18 (18) | 0.50 (0.24–1.06) | 0.063 |
| Adverse effects: | | | | |
| Discomfort | 3 (3) | 0 (0) | N/A | 0.25* |
| Pain in the lateral abdomen | 1 (1) | 0 (0) | N/A | 1.0* |

CI, confidence interval. N/A = not applicable. Values are numbers (percentages) unless stated otherwise. Chi-squared test was utilized unless stated otherwise.

*Fisher's exact test was utilized.

Table 3. Primary and secondary outcomes in the per-protocol analysis.

| Outcomes | No (%) of participants | | Relative risk (95% CI) | P value |
|---|---|---|---|---|
| | Lateral position group | Control group | | |
| | (n = 90) | (n = 73) | | |
| **Primary outcome** | | | | |
| Breech presentation at 37 weeks | 8 (9) | 15 (21) | 0.43 (0.19–0.96) | 0.020 |
| **Secondary outcomes** | | | | |
| Cephalic version 2 weeks after the start of the intervention | 68 (76) | 39 (53) | 0.53 (0.34–0.81) | 0.003 |
| Cephalic version 4 weeks after the start of the intervention | 74 (82) | 53 (73) | 0.65 (0.36–1.16) | 0.14 |
| Cephalic version 6 weeks after the start of the intervention | 76 (84) | 54 (74) | 0.60 (0.32–1.11) | 0.098 |
| Recurrence of breech presentation after cephalic version | 2/85 (2) | 9/64 (14) | 0.17 (0.037–0.75) | 0.010* |
| Overall cesarean delivery | 18 (20) | 30 (41) | 0.49 (0.30–0.80) | 0.003 |
| Cesarean delivery due to breech presentation | 7 (8) | 15 (21) | 0.38 (0.16–0.88) | 0.018 |
| Adverse effects: | | | | |
| Discomfort | 3 (3) | 0 (0) | N/A | 0.25* |
| Pain in the lateral abdomen | 1 (1) | 0 (0) | N/A | 1.0* |

CI, confidence interval. N/A = not applicable. Values are numbers (percentages) unless stated otherwise. Chi-squared test was utilized unless stated otherwise.

*Fisher's exact test was utilized.

## Subgroup analyses

In the subgroup analyses restricted to multiparous women, the intervention group had a significantly higher rate of cephalic version after 2 weeks (86% [31/36] versus 50% [19/38]); absolute risk difference, −15.6% [95% CI, −29.1% to −2.20%]; relative risk, 0.28 [95% CI, 0.02 to 0.67]; $p = 0.001$) and after 6 weeks (97% [35/36] versus 79% [30/38]; absolute risk difference, −18.3% [95% CI, −32.3% to −4.25%]; relative risk, 0.13 [95% CI, 0.017 to 1.00]; $p = 0.029$), and a lower rate of cesarean section (6% [2/36] versus 26% [10/38]); absolute risk difference, −20.8% [95% CI, −36.6% to −4.89%]; relative risk, 0.21 [95% CI, 0.05 to 0.90]; $p = 0.015$) than the control group (S1 Table). Conversely, the subgroup analyses restricted to primiparous mothers showed no significant differences between the two groups for all outcomes (S1 Table). In addition, we analyzed whether the intervention effects were the same or different between two subgroups, namely, primiparous and multiparous women. We used a logistic regression model to analyze the interaction between the intervention and the parity. S1 Table presents the obtained p-value of the coefficient measuring the interaction for each outcome. Only the rate of cephalic version after 2 weeks indicated statistical significance ($p = 0.014$).

## Adverse events

As adverse events in the lateral position instruction group, three cases of discomfort and one case of lateral abdominal pain were reported; nonetheless, all were resolved spontaneously, and the perinatal prognosis was good. In the control group, one experienced uterine rupture at delivery, which is a serious outcome. Other variables, namely, gestational age at delivery, birth weight, Apgar score, umbilical artery pH at birth, and blood loss amount, showed no significant differences between the two groups (Table 4).

## Discussion

We performed the largest randomized controlled trial of postural therapy for breech presentation to the best of our knowledge and revealed that the lateral position management

**Table 4. Obstetric outcomes.**

| Obstetric outcomes | Lateral position group | Control group |
|---|---|---|
| | (*n* = 100) | (*n* = 100) |
| Gestational age at delivery (weeks + days) | 39 + 1 (38 + 4–40 + 2) | 39 + 1 (38 + 1–40 + 1) |
| Delivery mode*: | | |
| Normal | 68 (68) | 53 (53) |
| Vacuum | 7 (7) | 11 (11) |
| Forceps | 2 (2) | 0 (0) |
| Cesarean | 23 (23) | 36 (36) |
| Vaginal breech birth | 0 (0) | 0 (0) |
| Birth weight | 3,047 (2,796–3,267) | 3,014 (2,774–3,243) |
| Apgar score: | | |
| 1 min | 8 (8–8) | 8 (8–8) |
| 5 min | 9 (9–9) | 9 (9–9) |
| Umbilical artery pH at birth | 7.305 (7.272–7.327) | 7.314 (7.273–7.334) |
| Bleeding amount during delivery (mL) | 450 (333–662) | 490 (330–860) |
| Perinatal complications*: | | |
| Fetal distress | 7 (7) | 12 (12) |
| Uterine inertia | 7 (7) | 10 (10) |
| Gestational diabetes mellitus | 6 (6) | 7 (7) |
| Hypertensive disorder of pregnancy | 4 (4) | 8 (8) |
| Atonic postpartum hemorrahge | 3 (3) | 3 (3) |
| Oligohydramnios | 4 (4) | 2 (2) |
| Failure to progress | 2 (2) | 2 (2) |
| Chorioamnionitis | 1 (1) | 2 (2) |
| Cervical laceration | 1 (1) | 2 (2) |
| Intrauterine growth restriction | 1 (1) | 1 (1) |
| Placental abruption | 2 (2) | 0 (0) |
| Postpartum mastitis | 2 (2) | 0 (0) |
| Postpartum depression | 1 (1) | 0 (0) |
| Uterine prolapse | 1 (1) | 0 (0) |
| Accreta placenta | 1 (1) | 0 (0) |
| Fourth perineal tear | 0 (0) | 1 (1) |
| Third perineal tear | 0 (0) | 1 (1) |
| Dural puncture | 0 (0) | 1 (1) |
| Uterine rupture | 0 (0) | 1 (1) |
| Meconium aspiration syndrome | 0 (0) | 1 (1) |

Missing data: Control group; bleeding amount during delivery (*n* = 1).

Values are median (interquartile range) unless stated otherwise.

There were no significant differences between the groups in any of the characteristics listed here except normal delivery and cesarean delivery.

*Values are numbers (percentages).

reduced breech presentation at the beginning of third trimester by about 1/2 compared with the control group at 37 gestational weeks, although this difference was not significant in the intention-to-treat analysis. The prespecified per-protocol analysis excluding crossover found the same direction of effect with statistical significance. The intention-to-treat analysis further showed significant differences between the two groups in head-first rates 2 weeks after the instruction and breech presentation recurrence rates after cephalic version. These findings are

clinically important because they indicate the possibility of reducing breech presentation and preventing breech recurrence just by lying down as long as the fetal back position is known.

## Comparison with other studies

Currently, no randomized controlled trials have tested the lateral position management for breech presentation, and all randomized controlled trials that have validated postural therapy for breech presentation have been related to the knee–chest position or the deep Trendelenburg position [11–14]. Conversely, one knee–chest position trial showed a higher cephalic version rate in the control group, and a meta-analysis found that the postural management for breech presentation has no sufficient supporting evidence and that further research is needed [11,12]. In our exploratory cohort study examining the effect of opposite side lateral position, the cephalic version rate at 37 weeks was 94% versus 77%, whereas that in the present study was 89% versus 81% in the intention-to-treat analysis [23]. Excluding protocol deviations, the per-protocol analysis showed a cephalic version rate of 91% versus 79%, similar to that in the prior study. Intervention at earlier weeks is generally considered unnecessary because of the high rate of spontaneous version, and most trials began at around 34 weeks; however, this study showed that interventions by 32 weeks, when the fetus is more likely to rotate than later, may be effective.

   This study found four cases of adverse events, all of which were minor and resolved spontaneously. Considering the safety of the lateral position originally reported in a meta-analysis, this study also confirms the safety of providing continuous lateral position instructions [24].

## Implications of findings

Most common postural managements for breech presentation, such as the knee–chest position or the deep Trendelenburg position, are based on the concept of lifting the pelvis to a higher position to float the fetus out of the pelvis and promote rotation, but holding the pelvis up for a long period of time is difficult. By contrast, the lateral position can be performed for a long time, because it is exactly the position in which a pregnant woman can usually sleep comfortably. In this study, the opposite side lateral position method significantly reduced breech presentation in the third trimester without lifting the pelvis after 2 weeks of instruction. This study is the first to experiment the concept of postural therapy to promote fetal self-rotation without lifting the pelvis, thereby offering a new concept for the treatment of breech presentation.

   In the same side lateral position (modification of the opposite side lateral position) performed after cephalic version, gravity is used to stabilize the fetus; however, its effectiveness has never been tested. In the present study, breech presentation recurrences were significantly fewer in the intervention group, indicating for the first time that the same side lateral position can prevent breech presentation. If the same side lateral position is proven effective, it would be inventive because no method to prevent breech presentation recurrence after cephalic version has existed. Early therapeutic intervention is not considered essential because of occasional breech presentation recurrence, but preventing breech recurrence by same side lateral position can make intervention from the early third trimester meaningful.

## Strengths and limitations of study

The strengths of this study are the stratified randomization by parity, which may have a strong influence on the outcome, the confirmation of the outcomes of all patients, few missing values, and the accurate and objective confirmation of fetal position by ultrasonography at each visit. The lateral position method is also cost-free, can be performed safely anywhere, and has

advantages from a medical economics perspective. In this study, ultrasonography was used to ensure objectivity in confirming the presentation and position of fetal back; however, if a skilled person can identify the fetal presentation and fetal back by abdominal palpation (Leopold maneuver), performing lateral position management may be possible in places where ultrasound equipment is not available. Currently, the external cephalic version is an established treatment method for breech presentation, but considering its risks and unsuccessful cases, the option of using the opposite side lateral position method as a premedication before the external cephalic version is considered [10,34]. Meanwhile, the same side lateral position could be applied as a maintenance therapy after a successful external cephalic version. In addition, rotating the fetus at earlier weeks of gestation and maintaining the cephalic presentation have the advantage of reassuring the mother early on.

The greatest limitation of this study is, as mentioned in the Statistical Analysis of Methods section, the presence of a potential lateral position group that happens to be in the same posture as the intervention group within the control group. Additionally, given that this study was open-label, participants in the control group might have intentionally learned and practiced the lateral positioning method secretly; in fact, three participants in the control group confessed that they had practiced the lateral positioning method. Moreover, the intervention group had 10 (10%) deviants in the intervention group. To address these issues caused by spontaneous and deliberate crossover, we performed a per-protocol analysis and found that lateral position management may be useful. Therefore, the lack of a significant difference in the primary outcome in the intention-to-treat analysis may be explained by the influence of these unplanned crossovers on effect size reduction, and further studies are needed to find a significant difference in the breech presentation at 37 gestational weeks in the intention-to-treat analysis; however, then single-center studies are no longer feasible, and large multicenter studies, preferably with clustering and other measures to eliminate any open-label bias, are needed. Finally, although our intention-to-treat analysis showed a significantly lower cesarean rate in the intervention group, it is necessary to take into account the fact that one prior cesarean section was incidentally included in the control group and that all patients who were in breech presentation at term underwent cesarean section, which is now common in Japan [5].

## Conclusions

The combination of the opposite side lateral and same side lateral position for breech presentation at 28–30 gestational weeks reduced breech presentation at 37 gestational weeks by approximately 1/2 compared with the control group, although not statistically significant in the intention-to-treat analysis. The prespecified per-protocol analysis showed the same tendency of the effectiveness with statistical significance. Regarding the secondary outcomes, the lateral positional group had a considerably high rate of cephalic version after 2 weeks of instruction and a low rate of breech recurrence. The direction of these effects is also promising, and this trial presents a new treatment concept for breech presentation, that is, correcting breech presentation and preventing breech recurrence simply by lying down based on fetal back. Although the application of the lateral position method before and after external cephalic version is assumed, further research considering the impact of unscheduled participants crossover may be warranted.

## Supporting information

**S1 Table. Primary and secondary outcomes in primiparous and multiparous women.**
(DOCX)

**S1 Protocol. Study protocol v1.0.**
(DOCX)

**S2 Protocol. Study protocol v2.1.**
(DOCX)

**S1 Plan. Statistical analysis Plan v1.0.**
(DOCX)

**S2 Plan. Statistical analysis Plan v1.2.**
(DOCX)

**S1 Data. De-identified data set of BRLT study.**
(XLSX)

**S1 CONSORT Checklist. CONSORT 2010 Checklist.**
(DOC)

**S1 CONSERVE Checklist. CONSERVE-CONSORT 2021 Checklist.**
(DOCX)

## Acknowledgments

We thank all the women who participated and all the staff involved in this study. The following individuals were involved in participant recruitment, ultrasound measurements, positional guidance, and outcome assessment: Ayako Takizawa, Aya Ohno, Honglian Shi, Asako Nagashima, Kei Sagawa, Nobuko Kato, Mayu Yamada, Yuki Kaito, Shingo Ogawa and Go Ichikawa. The following individuals were involved in the concept of this trial: Takehiko Fukami, Ikuno Kawabata, Masahiko Kato and Naofumi Okuda. Risa Shimmura made the allocation lists and sealed them. The authors would like to thank Enago for the English language review. Nobuhiko Taniai was responsible for auditing this trial.

## Author contributions

**Conceptualization:** Hiroki Shinmura, Youhei Tsunoda, Takashi Matsushima, Shunji Suzuki.

**Data curation:** Youhei Tsunoda, Takashi Matsushima.

**Formal analysis:** Youhei Tsunoda, Shunji Suzuki.

**Investigation:** Hiroki Shinmura, Ryuhei Kurashina, Asako Watanabe, Eika Harigane, Nozomi Ouchi.

**Methodology:** Hiroki Shinmura.

**Project administration:** Hiroki Shinmura, Ryuhei Kurashina, Asako Watanabe, Eika Harigane, Nozomi Ouchi.

**Software:** Hiroki Shinmura.

**Supervision:** Takashi Matsushima, Shunji Suzuki.

**Validation:** Hiroki Shinmura, Youhei Tsunoda, Takashi Matsushima, Shunji Suzuki.

**Visualization:** Hiroki Shinmura.

**Writing – original draft:** Hiroki Shinmura.

**Writing – review & editing:** Youhei Tsunoda, Takashi Matsushima, Ryuhei Kurashina, Asako Watanabe, Eika Harigane, Nozomi Ouchi, Shunji Suzuki.

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
