## [Editor Report · Decision Letter 0]

17 Jun 2024

Dear Dr Shinmura, 

Thank you for submitting your manuscript entitled "Reduction of breech presentation and prevention of breech recurrence just by lying down in lateral position based on fetal back: a randomized controlled trial (BRLT study)" for consideration by PLOS Medicine.

Your manuscript has now been evaluated by the PLOS Medicine editorial staff and I am writing to let you know that we would like to send your submission out for external peer review. Please provide copies of the original protocol (as approved by the ethics committee) and the original statistical analysis plan as supplementary files in your resubmission. 

Before we can send your manuscript to reviewers, we also need you to complete your submission by providing the metadata that is required for full assessment. To this end, please login to Editorial Manager where you will find the paper in the 'Submissions Needing Revisions' folder on your homepage. Please click 'Revise Submission' from the Action Links and complete all additional questions in the submission questionnaire.

Please re-submit your manuscript within two working days, i.e. by Jun 19 2024 11:59PM.

Feel free to email me at lgaynor@plos.org if you have any queries relating to your submission.

Kind regards,

Louise Gaynor-Brook, MBBS PhD

Senior Editor

PLOS Medicine

---

## [Decision Letter · Decision Letter 1]

20 Aug 2024

Dear Dr Shinmura,

Many thanks for submitting your manuscript "Reduction of breech presentation and prevention of breech recurrence just by lying down in lateral position based on fetal back: a randomized controlled trial (BRLT study)" (PMEDICINE-D-24-01899R1) to PLOS Medicine. The paper has been reviewed by subject experts and a statistician; their comments are included below and can also be accessed here: [LINK]

After discussing the paper with the editorial team and an academic editor with relevant expertise, I'm pleased to invite you to revise the paper in response to the reviewers' comments. We plan to send the revised paper to some or all of the original reviewers, and we cannot provide any guarantees at this stage regarding publication.

We ask that you submit your revision by Sep 10 2024 11:59PM. However, if this deadline is not feasible, please contact me by email, and we can discuss a suitable alternative.

Don't hesitate to contact me directly with any questions (lgaynor@plos.org). 

Best regards, 

Louise 

Louise Gaynor-Brook, MBBS PhD 

Senior Editor

PLOS Medicine

lgaynor@plos.org

Comments from the editors:

We note some discrepancies between the original protocol and this report of the findings from the trial. Specifically, these relate to:

1) Primary and secondary outcomes

We note that the primary endpoint in protocol was specified as the rate of head conversion 2 weeks after teaching postural therapy, whereas the primary endpoint in manuscript was % of breech presentation cases at 37 weeks gestation. It appears that primary and secondary outcomes with relation to % breech at 37 gestational weeks and head presentation at 2 weeks after the intervention have been switched.

Please provide details of any protocol amendments to justify changes to the primary outcomes of the trial. Please provide documentary evidence that the changes were approved by a steering committee / IRB.

2) Power calculation

We note that the predicted conversion rates changed in the power calculation. In the protocol, it was assumed that there would be 80% and 60% conversion in intervention and control arms respectively. In the manuscript, it is assumed that there would be 94% and 80% conversion in intervention and control arms respectively. Originally, n=165 were required per group. In the manuscript, n=100 are required per arm.

Please provide justification for the changes made to the above assumptions and sample size. If amendments to the trial protocol and SAP were made after the original versions of these documents were finalised, please provide copies of the final versions of the protocol and SAP as a supplementary file alongside your revised manuscript. If the SAP differs from the protocol, please justify any changes. 

In addition, in line with comments from the reviewers and the Academic Editor, please provide a copy of the original (raw) data in your revised version. 

Comments from the reviewers: 

Reviewer #1: Alex McConnachie, Statistical Review

The paper by Shinmura et al presents the results of a randomised trial of a postural therapy intervention to prevent breech presentation, in 200 women from Japan. This review looks at the use of statistics in the paper.

There are several good things about the statistical aspects in this paper. The sample size justification is quite clear, and the calculation is repeatable. The main analyses are by intention to treat, and the methods themselves are nice and simple (Chi-square test, or Fisher Test), with results presented as relative risks, with confidence intervals and p-values, which are easy to understand.

However, there are a few areas where things could be improved, one of which would make the results clearer, and a couple that would make them more robust (in my opinion).

First, the primary outcome data are presented as the percentage of women with a breech presentation at 37 weeks being reduced in the intervention arm: 11% vs 19%. Since the risk in the intervention arm is lower, this appears as a RR<1, namely 0.91. The problem is that 11/19 = 0.58. It appears that the authors have calculated the RR as the risk of not having the primary outcome in the control arm (81%) divided by the risk of not having the primary outcome in the intervention arm (89%). This seems to happen for many of the outcomes, but is not entirely consistent. In general, when talking of relative "risk", that implies looking at the probability of the adverse outcome. Also, when comparing treatment groups, one normally reports the risk in the intervention arm divided by the risk in the control arm, so that if the risk is reduced, the RR is less than 1.

Secondly, the authors present per-protocol analyses. This is commonly done, but is not without problems, since the population for analysis is based on participants' behaviour after being randomised, giving a biased estimate of the effect of the intervention. Nowadays, there are better methods, such as Complier Average Causal Effects (CACE) methods, which can estimate the effect of the intervention, allowing for non-compliance in the intervention arm, as well as a proportion of the control arm receiving the intervention. If possible, it would be good to provide CACE estimates of intervention effects. Since this was not pre-specified, it would be fine to put this into a supplement.

Another thing that could be a little better is the subgroup analysis of primiparous vs multiparous women. Currently, the analysis is done separately in the two subgroups, and a significant result is shown for multiparous women, but not for the primiparous group. This is interesting, but on its own, does not allow any conclusions to be drawn. It is better to do an analysis that formally tests whether the intervention effects in the two subgroups are the same or different. For example, a logistic regression model with an interaction between parity and the intervention would achieve this. Only if this interaction test shows some evidence of a difference does it become reasonable to interpret the separate intervention effect estimates. Since the study was designed to detect an overall intervention effect, any interaction test will be underpowered, but this would be a more robust approach.

Other points:

Line 290-291 states that one women was excluded from the PP dataset due to a planned caesarean at 38 weeks. Why, if the primary outcome was measured at 37 weeks?

The section on subgroups analyses only reports p-values. As stated above, an interaction test of some sort should be included, but when presenting the results within subgroups, the intervention effect estimates and confidence intervals should be reported. Also, would it be possible to show the subgroup analyses in a forest plot? Maybe it wouldn't work, because this is usually done when there is one (primary) outcome and several subgroup analyses; here we have multiple outcomes, looked at in the same subgroups.

The methods section says that results are reported as relative risks, and absolute risk differences (as they should, according to CONSORT), but I do not recall seeing the absolute risk differences presented or discussed anywhere.

Reviewer #2: Thank you for asking me to review this paper.

Overall, my main discomfort with the paper is the authors' continued use of the terms 'validated.' The study did not demonstrate a statistically significant difference in the primary outcome. It has provided data that suggest further research is warranted. But much more research is warranted before this can be recommended. In many settings, it would require introducing interventions (eg. scans, advice) that are not currently used at earlier gestations.

This RCT has been performed on a sample of women having routine ultrasound scanning at 28 to 30 weeks. Its applicability outside of Japan is therefore contained to settings which perform routine USS at 28 to 30 weeks, and who begin a breech care pathway at this point if identified. This would not include the UK, where routine ultrasound scanning is not performed at this time, and where the usual NHS breech care pathway does not begin until 36 weeks. There is very little likelihood that an intervention requiring an USS every two weeks from 30 weeks would be implemented.

Caesarean birth is also performed at 39 weeks in the UK and most countries, due to the preference for fetal lung maturity. Again, this brings into question the applicability to international practice.

Having read the article, I find the terms 'lateral position' and 'reverse lateral position' difficult to understand and counter-intuitive. For example, in reverse lateral position, the mother is lying on the SAME side where the baby's back lies. In lateral position, it is the OPPOSITE side as the baby's back. These terms seem the opposite of what I would assume. 

To English language speakers, the terms 'same side lateral position,' meaning same side as the baby's back, and 'opposite side lateral position,' meaning opposite side as the baby's back, may make more sense. I would encourage the authors to reach out to an international audience and sense-check these terms.

Abstract:

"Few studies have validated lateral position management for breech presentation …" This assumes that readers know what lateral position management means. We don't. Try using plain English to describe what you are doing, from the start.

The concluding statements are not warranted. The study did not demonstrate a statistically significant difference in the primary outcome, breech presentation at 37 weeks. The direction of effect is promising, but this can only be considered a pilot study, not a basis for clinical recommendations. Also, the study did not aim to assess the effect of 'reverse lateral position' as a maintenance therapy. Again, recommendations are not warranted.

p.4, line 58. 'Vaginal delivery in breech presentation is associated with perinatal morbidity and mortality.' - This statement needs a lot more specificity. Pregnancy, and any type of birth, is associated with perinatal morbidity and mortality.

Line 59 - Caesarean section was recommended by whom? ACOG? This isn't the most recent guidance. Is it the guidance followed in Japan?

Line 61 - 'The only established treatment for breech presentation is external cephalic version.' What does this mean? External cephalic version is not the only way to 'treat' breech presentation. There are many methods of delivering breech babies vaginally as well.

Listing potential 'treatments' for breech presentation before 36 weeks and saying their effectiveness 'remains controversial,' is vague. This paper should include a scientific summary of the availably literature.

Line 68 - "In 1812, Wigand described the lateral position as effective when the fetal head descended from the transverse position with the fetal back on the fundal uterine side." - I do not understand what this means at all. What does 'lateral position' mean - which lateral position? What does 'fundal uterine side' mean? This paper requires clearer explanations.

I have scrolled to the bottom and identified the figures with the lateral positions. These need to be used very early on in the paper, so that readers have a common understanding of terms used. 'Lateral position' and 'reverse lateral position' should be defined, with figures, from the very beginning.

I would like to openly declare my biases: I research techniques to make vaginal breech birth safer. I have been working in breech clinics and researching breech care for 14 years. I am aware of how much time and effort it takes to 1) identify breech presentations; 2) counsel women about even position changes; and 3) monitor outcomes. There are economic costs to all of this, both to services and to women and families, due to needing to present for healthcare (societal costs).

Therefore, before any recommendations can be made, definitive trials need to be conducted, and these need to have an economic component.

I would encourage the authors to consider becoming aware of recent developments in vaginal breech delivery as well, and whether this impacts whether such early, frequent assessment and intervention in breech presentation is cost- and clinically effective.

Dr Shawn Walker

Midwife, University College Hospital London

Honorary Senior Research Fellow, King's College London

Reviewer #3: Thank you for allowing me to review PMEDICINE-D-24-01899R1 entitled Reduction of breech presentation and prevention of breech recurrence just by lying down in lateral position based on fetal back: a randomized controlled trial (BRLT study).I have read the paper several times. Congratulations on this effort.

The study is well executed; trial registration in place and protocol published.

One suggestion is to have a native speaker with knowledge of clinical trials to edit the manuscript. While there are 

I miss table with obstetric outcomes: gestational age at delivery, delivery mode, birth weight,

Apgar scores at 1 and 5 minutes, umbilical artery pH at birth, bleeding amount during

delivery, and perinatal complications. 

This should all be tabulated.

I would like to see the raw data underlying the paper.

Minor issue: Inclusion criteria: " gestational age between 28+0 and 30+0 gestational weeks". Results: The median number of gestational weeks at randomization was 28 weeks for all groups. Table says 28+6.

---

* Please upload any figures associated with your paper as individual TIF or EPS files with 300dpi resolution at resubmission; please read our figure guidelines for more information on our requirements: http://journals.plos.org/plosmedicine/s/figures. While revising your submission, please upload your figure files to the PACE digital diagnostic tool, https://pacev2.apexcovantage.com/. PACE helps ensure that figures meet PLOS requirements. To use PACE, you must first register as a user. Then, login and navigate to the UPLOAD tab, where you will find detailed instructions on how to use the tool. If you encounter any issues or have any questions when using PACE, please email us at PLOSMedicine@plos.org.

FIGURES AND TABLES

SUPPLEMENTARY MATERIAL

REFERENCES

STUDY TYPE-SPECIFIC REQUESTS: RCTs 

* PLOS Medicine requires that all trials be prospectively registered in one of registries recognized by WHO. Please ensure that study registration details are included in the Methods section.

* Please structure the Methods section using the following sub-headings: Study design and participants, Randomization and masking, Procedures, Outcomes, Statistical analysis.

* The primary and secondary outcomes appear to differ between the submitted manuscript and the protocol [and/or trial registry]. Please clarify and explain all discrepancies between the paper and protocol. If the outcomes were not prespecified in the protocol, please define them in the Methods (Outcomes section) as post hoc and explain why they were added. Post-hoc comparisons should be presented as hypothesis generating rather than conclusive.

* Please ensure that all prespecified outcomes (primary, secondary, and exploratory) are listed in the Methods/Outcomes section and indicate whether there are outcomes that are not presented in the current report.

* Please specify the dates (Month Day, Year) during which study enrollment and follow up occurred.

* Please include absolute numbers wherever you report percentages; eg, n/N (%)

* Please present the safety data for the study including numbers of specific events and whether or not adverse events are thought to be related to treatment. AEs should be reported in the abstract, per CONSORT and CONSORT-Harms.

* Please complete the CONSORT checklist (https://www.equator-network.org/reporting-guidelines/consort/) and ensure that all components of CONSORT are present in the manuscript, including how randomization was performed, allocation concealment, blinding of intervention, definition of lost to follow-up, power statement. When completing the checklist, please use section and paragraph numbers, rather than page numbers.

* Please report your abstract according to CONSORT for abstracts, following the PLOS Medicine abstract structure (Background, Methods and Findings, Conclusions) https://www.equator-network.org/reporting-guidelines/consort-abstracts/

* If your trial had to undergo important modifications in response to extenuating circumstances, please complete the CONSERVE-CONSORT checklist and provide in your Supporting Information; (https://www.equator-network.org/reporting-guidelines/guidelines-for-reporting-trial-protocols-and-completed-trials-modified-due-to-the-covid-19-pandemic-and-other-extenuating-circumstances-the-conserve-2021-statement/). When completing the checklist, please use section and paragraph numbers, rather than page numbers.

* In keeping with our commitment to Open Science, please include the study protocol document and analysis plan (including any amendments) as Supporting Information to be published with the manuscript if accepted.

* Please note that PLOS Medicine requires prospective, public registration of a data sharing plan (as part of mandatory clinical trials registration) for all clinical trials that began enrollment on or after January 1, 2019, in accordance with ICMJE requirements.

---

## [Decision Letter · Decision Letter 2]

11 Nov 2024

Dear Dr Shinmura,

Many thanks for submitting your manuscript "Reduction of breech presentation and prevention of breech recurrence just by lying down in lateral position based on fetal back: a randomized controlled trial (BRLT study)" (PMEDICINE-D-24-01899R2) to PLOS Medicine. The paper has been re-reviewed by subject experts and a statistician; their comments are included below and can also be accessed here: [LINK]

After discussing the paper with the editorial team and an academic editor with relevant expertise, I'm pleased to invite you to revise the paper in response to the reviewers' comments. We plan to send the revised paper to some or all of the original reviewers, and we cannot provide any guarantees at this stage regarding publication.

We ask that you submit your revision by Dec 02 2024 11:59PM. However, if this deadline is not feasible, please contact me by email, and we can discuss a suitable alternative.

Don't hesitate to contact me directly with any questions (lgaynor@plos.org). 

Best regards, 

Louise 

Louise Gaynor-Brook, MBBS PhD 

Senior Editor

PLOS Medicine

lgaynor@plos.org

Comments from the editors:

For full transparency, please make very clear which, when and why protocol changes were made, and add these to the main manuscript text (Methods section). 

It appears as though a protocol amendment was made to switch primary and secondary outcomes (approved by the IRB) 16 months into recruitment; please comment on whether this affected the informed consent provided by participants prior to the protocol amendment. How many/what proportion of the participants in the trial had been recruited at the time of the protocol amendment? 

Please note that the raw data as provided contain potentially identifiable information and is unsuitable for publication in its current form. Please provide fully de-identified data underlying the specific results. 

Comments from the reviewers: 

Reviewer #1: Alex McConnachie, Statistical Review

I thank the authors for their consideration of my original comments.

I think the reporting of risks and risk ratios is generally improved, though the wording of the results of cephalic version at 2 weeks is a little awkward, since the outcome is a positive event, but the RR is given for the adverse outcome of the event not happening. Perhaps, if the paper is accepted, the journal's editorial team can help find a suitable form of words here.

I applaud the authors' attempt at a CACE analysis, but I am not sure it is done completely correctly. Perhaps it would be better to leave this out.

For the subgroup analysis, I think I may have been misunderstood. The authors have fitted a logistic regression model to estimate the effect of the intervention with adjustment for parity. This is not an unreasonable thing to do, but not quite what I meant. What I was suggesting is fitting a model with an interaction between the intervention and parity. 

Let x1 be defined as 1 for those randomised to the intervention and 0 for those randomised to control

Let x2 be defined as 1 for multiparous women and 0 for nulliparous women

Then, let x3 be defined as x1 multiplied by x2. x3 is then said to measure the interaction between the intervention (x1) and parity (x2).

Then, fit a logistic regression model with x1, x2, and x3 as predictors. The paper should report the p-value for the x3 coefficient. If this is significant, it indicates that the intervention effect in multiparous women is different to the intervention effect in nulliparous women.

Finally, I was not convinced by the reasoning for excluding the woman with a planned caesarean from the PP analyses. I accept that the foetal position may be clinically irrelevant if a caesarean is planned, but from the perspective of the trial, the primary outcome at 37 weeks was still measurable, and provides valid information about the effectiveness of the intervention.

Reviewer #2: Thank you for asking me to review this revision of this manuscript.

I am much more comfortable with the language the authors use to describe their intervention, which would enable it to be more easily replicated in other tests.

My co-reviewers have addressed trial management and statistical concerns thoroughly. I will leave it to their judgement as to whether the revisions adequately address their concerns.

Dr Shawn Walker

Midwife, University College London Hospitals NHS Trust

Visiting Senior Research Fellow, King's College London

Reviewer #3: The authors have responded well on the comments of the reviewers. I recommend publication.

I would suggest to publish the raw de-identified data with the paper.

---

* Please upload any figures associated with your paper as individual TIF or EPS files with 300dpi resolution at resubmission; please read our figure guidelines for more information on our requirements: http://journals.plos.org/plosmedicine/s/figures. While revising your submission, please upload your figure files to the PACE digital diagnostic tool, https://pacev2.apexcovantage.com/. PACE helps ensure that figures meet PLOS requirements. To use PACE, you must first register as a user. Then, login and navigate to the UPLOAD tab, where you will find detailed instructions on how to use the tool. If you encounter any issues or have any questions when using PACE, please email us at PLOSMedicine@plos.org.

FIGURES AND TABLES

SUPPLEMENTARY MATERIAL

REFERENCES

RCTs 

* PLOS Medicine requires that all trials be prospectively registered in one of registries recognized by WHO. Please ensure that study registration details are included in the Methods section.

* Please structure the Methods section using the following sub-headings: Study design and participants, Randomization and masking, Procedures, Outcomes, Statistical analysis.

* Please clarify and explain all discrepancies between the paper and protocol in the main manuscript text. If the outcomes were not prespecified in the protocol, please define them in the Methods (Outcomes section) as post hoc and explain why they were added. Post-hoc comparisons should be presented as hypothesis generating rather than conclusive.

* Please ensure that all prespecified outcomes (primary, secondary, and exploratory) are listed in the Methods/Outcomes section and indicate whether there are outcomes that are not presented in the current report.

* Please specify the dates (Month Day, Year) during which study enrollment and follow up occurred.

* Please include absolute numbers wherever you report percentages; eg, n/N (%)

* Please present the safety data for the study including numbers of specific events and whether or not adverse events are thought to be related to treatment. AEs should be reported in the abstract, per CONSORT and CONSORT-Harms.

* Please complete the CONSORT checklist (https://www.equator-network.org/reporting-guidelines/consort/) and ensure that all components of CONSORT are present in the manuscript, including how randomization was performed, allocation concealment, blinding of intervention, definition of lost to follow-up, power statement. When completing the checklist, please use section and paragraph numbers, rather than page numbers.

* Please report your abstract according to CONSORT for abstracts, following the PLOS Medicine abstract structure (Background, Methods and Findings, Conclusions) https://www.equator-network.org/reporting-guidelines/consort-abstracts/

* If your trial had to undergo important modifications in response to extenuating circumstances, please complete the CONSERVE-CONSORT checklist and provide in your Supporting Information; (https://www.equator-network.org/reporting-guidelines/guidelines-for-reporting-trial-protocols-and-completed-trials-modified-due-to-the-covid-19-pandemic-and-other-extenuating-circumstances-the-conserve-2021-statement/). When completing the checklist, please use section and paragraph numbers, rather than page numbers.

* In keeping with our commitment to Open Science, please include the study protocol document and analysis plan (including any amendments) as Supporting Information to be published with the manuscript if accepted.

* Please note that PLOS Medicine requires prospective, public registration of a data sharing plan (as part of mandatory clinical trials registration) for all clinical trials that began enrollment on or after January 1, 2019, in accordance with ICMJE requirements.

---

## [Decision Letter · Decision Letter 3]

24 Jan 2025

Dear Dr. Shinmura,

Thank you very much for re-submitting your manuscript "Reduction of breech presentation and prevention of breech recurrence just by lying down in lateral position based on fetal back: A randomized controlled trial (BRLT study)" (PMEDICINE-D-24-01899R3) for review by PLOS Medicine. I am writing in the place of my colleague Dr. Louise Gaynor-Brook, who is presently out of the office.

I have discussed the paper with my colleagues and it was also seen again by the statistical reviewer, who requests some additional analysis. Provided the remaining referee, editorial and production issues are dealt with, we are planning to accept the paper for publication in the journal.

The remaining issues that need to be addressed are listed at the end of this email. Comments on the abstract are in the attached file. Any accompanying reviewer attachments can be seen via the link below. Please take these into account before resubmitting your manuscript:

[LINK]

We look forward to receiving the revised manuscript by Jan 31 2025 11:59PM.   

Sincerely,

Alison Farrell, PhD

Senior Editor

PLOS Medicine

afarrell@plos.org

Requests from Editors:

* Title: Please confirm that your title complies with to PLOS Medicine's style. Your title must be nondeclarative and not a question. It should begin with main concept if possible. "Effect of" should be used only if causality can be inferred, i.e., for an RCT. Please place the study design ("A randomized controlled trial," "A retrospective study," "A modelling study," etc.) in the subtitle (ie, after a colon. We suggest: Evaluation of postural therapy on breech presentation and breech recurrence (BRLT study): An open-label, randomized controlled trial

*Abstract: Please include the clinical trial registry number in the abstract.

*Figures: *Figure 1 must be the study flow chart. Please reorganize the figures and figure call-outs in the text accordingly.

*Main text: There are numerous language issues thought the text. Please seek the assistance of a native English language speaker or editor to revise the full manuscript. Please also review your manuscript and edit to ensure compliance with our inclusive language requirements https://journals.plos.org/plosmedicine/s/human-subjects-research#loc-categorization

*Methods: Please include first and last dates of participant enrolment in the Methods section. Please identify and cite in the Methods the institutional review board(s) that approved the study, separate to the ethics committee.

Please also amend the reference to informed consent. As stated on line 179, the wording “All participants were provided written informed consent before randomization” indicates that the participants were provided with informed consent. Please delete ‘were’ to indicate that the participants themselves provided the consent.

Please make sure all of the secondary endpoints are described and whether they are all reported. Please signpost any exploratory endpoints or post hoc analyses.

In view of the lack of statistically significant difference in the primary endpoint in this study, the change in study protocol, and the crossover events, please ensure that your representation of the study results and conclusions is appropriately tempered.

*Funding: please state the study funders and their role in the study design, execution and analysis.The funding statement should include: specific grant numbers, initials of authors who received each award, URLs to sponsors’ websites. Also, please state whether any sponsors or funders (other than the named authors) played any role in study design, data collection and analysis, the decision to publish, or preparation of the manuscript. If they had no role in the research, include this sentence: “The funders had no role in study design, data collection and analysis, decision to publish, or preparation of the manuscript.

*Author contributions: Please clarify by what is meant by “HS invented the same side lateral position” . Do you mean that HS invented this approach to management of breech position? Please restate.

*Data Availability: Please revise the Data Availability Statement as we do not permit the use of “available upon reasonable request”. For each data source used in your study: 

* PLOS Medicine requires that the de-identified data underlying the specific results in a published article be made available, without restrictions on access, in a public repository or as Supporting Information at the time of article publication, provided it is legal and ethical to do so. Please see the policy at 

http://journals.plos.org/plosmedicine/s/data-availability

and FAQs at 

http://journals.plos.org/plosmedicine/s/data-availability#loc-faqs-for-data-policy

Please include the deidentified data in the manuscript, as requested by the reviewers.

* As your trial had to undergo important modifications in response to extenuating circumstances, please complete the CONSERVE-CONSORT checklist and provide in your Supporting Information.

When completing the checklist, please use section and paragraph numbers, rather than page numbers. When completing the CONSORT checklist, please use section and paragraph numbers rather than page numbers.

*Protocol: Please provide an English language only protocol.

Comments from Reviewers:

Reviewer #1: Alex McConnachie, Statistical Review

I thank the authors once again for responding to my comments.

Mostly these are fine. The authors have done an interaction test as part of their subgroup analyses, but unfortunately did not state which outcome it related to. I assume it is the primary outcome, but it is not clear. However, this should be done for all outcomes. Currently, the paper highlights certain significant results in the multiparous subgroup, in contrast to no significant results in the primiparous group.

In general, when reporting subgroup analyses, it is not enough to look at the p-values, and to comment on the significance or otherwise of associations within subgroups. Only by testing the interactions do you have evidence of whether the associations are actually different in one subgroup compared to the other.

If possible, I would merge Tables S1 and S2 into a single table (probably in landscape format), and add a column at the end showing the interaction p-values - one for each outcome. This would at least allow the reader to fully assess the results reported.

[LINK]

---

## [Editor Report · Decision Letter 4]

5 Feb 2025

Dear Dr Shinmura, 

On behalf of my colleagues and the Academic Editor, Andrew Shennan, I am pleased to inform you that we have agreed to publish your manuscript "Evaluation of postural therapy using lateral position according to fetal back orientation on breech presentation and breech recurrence (BRLT study): An open-label, randomized controlled trial" (PMEDICINE-D-24-01899R4) in PLOS Medicine.

PRESS

Sincerely, 

Rebecca Kirk 

Senior Editor 

PLOS Medicine